# Teaching machine learning to energy engineers through end-to-end decision making

**Hussain Kazmi** [1]

## Abstract

Advances in machine learning and data science hold the potential to greatly optimize the overall energy sector, and prevent the worst outcomes of anthropogenic climate change. However, despite the urgent need for trained energy data scientists and the presence of a number of technically challenging issues that need to be tackled, the sector continues to suffer from a personnel shortage and remains mired in outdated technology. In many programs, energy engineers continue to graduate without even rudimentary programming skills, let alone knowledge of data science. This paper highlights key findings from an introductory course on machine learning and optimization designed specifically for energy engineering students. The course employs a number of teaching aids, which we hope will be useful for the broader community as well. The course was developed in a pan-European setting, supported by four different European universities as part of a broader roadmap to overhaul energy education.

## 1. Introduction

Despite its importance, energy engineers typically do not get even an introductory course to data science or machine learning. This oversight is especially tedious since graduating students and fresh graduates have to learn these concepts in an adhoc manner. The omission is also becoming glaringly obvious with the increasing amounts of energy-related data being collected, thanks to IoT devices and smart meters etc. which are enabling countless use cases on both the building (Kazmi and Driesen, 2020) and the grid scale (Zhang et al., 2018). Recent years have also seen technological companies expand their energy departments to ensure compliance with ambitious decarbonization targets. However, as things stand, most energy companies and engineers are ill-equipped to cope with this data, much less create additional value from it.

To address this shortcoming, EIT InnoEnergy, a body of the European Union, initiated a working group, constituted of members from multiple universities (KU Leuven, KTH, UPC and Grenoble INP) in four different European countries (Belgium, Sweden, Spain and France) in 2019. The multi-year mandate of the working group was to harmonize data science education across the participating universities, reduce replication work in course design and creation, create grounds for broader collaboration, and develop a long-term roadmap for data science education in energy programs in European universities.

A note on terminology is relevant here. The working group converged to the use of data science as an umbrella term that incorporates the entire data pipeline (including machine learning, but also other closely related topics including data acquisition, exploratory data analysis, optimal decision making and ethics etc.). Furthermore, due to the specific nature of the target audience (i.e. energy engineers), an emphasis was placed on case-based teaching. This was intended to help students better understand how the algorithms are applied in practice, as well as what problems do they solve concretely.

One of the first deliverables arising from the working group's activies was an introductory course on data science for energy engineers that was delivered in a pilot run to students at KU Leuven. This was followed by a second and third (virtual) run during the summer of 2020 and 2021 respectively to students from over ten European universities. In this paper, we highlight some of the key findings in designing such a course from scratch. We also discuss lessons we learned while teaching machine learning to engineering students with diverse backgrounds.

## 2. Course audience

The course, titled 'Data science for energy engineers', is intended for graduate level (MS or early PhD) engineering students. The audience is then, in many ways, much less diverse than students enrolled in a typical introductory data

---
[*]Equal contribution [1]Dept. of Electrical Engineering, KU Leuven, Belgium. Correspondence to: Hussain Kazmi <hussain-syed.kazmi@kuleuven.be>.

*Proceedings of the $2^{nd}$ Teaching in Machine Learning Workshop*, PMLR, 2021. Copyright 2021 by the author(s).

science or artificial intelligence course. An overwhelming majority, if not all, of the students are engineering students, belonging mostly to the electrical, mechanical or energy departments. However, within these students, there is still considerable diversity owing to two key factors. Firstly, there are typically a number of differences between electrical and mechanical engineering curricula. Electrical engineers tend to have much more exposure to programming and fields closely related to machine learning such as signal processing. Secondly, individual student background also contributes considerably to diversity. For instance, high school students in many countries (can) already study programming, while in many cases it is possible that even electrical engineering graduates may make it through having written only a single 'hello world' program in Matlab. Reflecting our experience, this lack of programming knowledge has been identified as one of two main barriers for students learning machine learning and data science (Sulmont et al., 2019).

Consequently, the course was designed in a way that it would cater to the needs of a variety of energy engineering students, irrespective of their background. While the course, at 3 ECTS, is too short to include a detailed introduction to programming, students were provided with relevant documentation and material. Furthermore, the level of programming and software engineering knowledge required to complete the course was intentionally kept low (i.e. concepts such as object oriented programming, repository control etc. were alluded to, but not dealt with).

## 3. Intended learning outcomes

The course was crafted in a way that a number of concrete learning outcomes could be realized, while simultaneously introducing students to the end-to-end pipeline of data-driven projects ranging from data acquisition and preprocessing, to modelling and inference, to actionable decision making. Keeping in mind course participant backgrounds, these learning outcomes can be formalized as:

1. Students should be able to load, as well as visualize and understand various energy datasets by performing exploratory data analysis on them.

2. Students should be able to understand core machine learning algorithms for modelling and forecasting, and how (and when) to apply these in practical settings.

3. Students should be able to formulate and solve optimization problems to perform optimal control and design in a number of different energy related settings.

Additionally, students should be able to generalize from individual algorithms, and understand the complexity behind real world deployment.

## 4. Course content and design

To realize these learning objectives, we designed the course as a series of five lectures, accompanied by four practice sessions based on Jupyter notebooks. The individual lectures cover (1) introduction to energy data science and practical use cases, (2) exploratory data analysis, (3) modelling and forecasting using machine learning, (4) optimal decision making, and (5) advanced concepts in machine learning and optimization. The entire course is delivered using a singular dataset on electricity demand and prices, which allows students to work on it in an end-to-end manner. In this section, we highlight two different aids in course design. In the next section, we discuss three teaching aids in course delivery.

### 4.1. Teaching with end-to-end use cases

Rather than introducing students to machine learning with toy datasets such as the Iris or MNIST dataset, we designed the course using domain specific examples in the form of a coherent use case on energy demand response. This enabled us to discuss the entire life cycle of a real-world project in practice, of which building a machine learning or optimization model is just one step. Towards this end, we discuss the real world case of an energy prosumer (i.e. a residential household with local storage and/or generation) interested in optimizing their energy demand and generation using electric batteries. The course discusses a number of optimization objectives, ranging from performing arbitrage (i.e. using an optimal controller that charges the battery when electricity is cheaper or less carbon-intensive, and vice versa) to peak shaving (i.e. reducing the maximum power demand on the grid) and maximum self-consumption of local solar generation. Additional constraints can be introduced here, based on user behaviour and grid conditions. We also take care to emphasize that the same algorithms can be used to achieve a variety of objectives, ranging from energy optimization to cost optimization to emissions optimization.

This case study requires students to create forecasts for user energy demand. To do this, students are introduced to machine learning forecasting models, but emphasis is placed on benchmarking them using simpler methods (both naive and simple time series models). The temporal structure of the problem also allows us to introduce complex, real-world challenges such as anomalous data and concept drift etc. Another benefit of using a dataset that spans an entire year is that it reinforces the concept of statistical significance. While a naive forecast and/or controller may outperform a more sophisticated counterpart on a given day, the superior algorithm should outperform over a longer period of time.

Posing the problem in such a relatable manner also allows students to easily see the real-world costs of prediction errors from machine learning models, and whether complex machine learning algorithms actually improve real-world

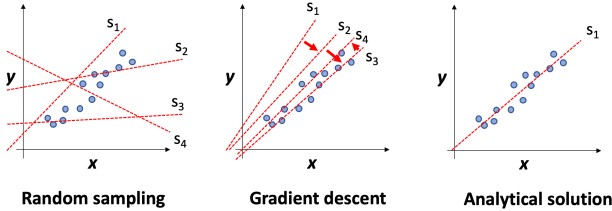

*Figure 1.* Introducing structure through understandable examples, both in machine learning and optimal decision making

results when compared to simpler baselines. Finally, it is important to note that the methodology is generalizable across case studies. For instance, in a partner university in the working group, a largely similar approach was followed with detecting wind turbine failures, where the students were again asked to make the link between prediction errors and real world costs.

### 4.2. Teaching using algorithmic structure

A second recurring theme throughout the course was showing students the existence of different algorithmic archetypes that can be used to solve the same problem (rather than superficially introducing a large number of machine learning algorithms). The archetypes were chosen to illustrate how structure inherent in problems can reduce computational complexity, while simultaneously improving the quality of the solution.

For instance, in *machine learning* it is possible to fit a regression curve to a training dataset using a variety of approaches. To demonstrate this, we start off with a conventional scatter plot (where the independent variable is on the x-axis and the dependent variable is on the y-axis). After this, we show students that it is possible to fit infinitely many (linear) curves through the point cloud. These curves can be quickly evaluated, but it should already be obvious to the students by this point that the prospects of attempting a large number of solutions to determine the best one is terribly wasteful. As the next step, we discuss a variety of gradient-based and gradient-free algorithms which can be used to arrive at the optimal curve in far fewer iterations. From here, we informally introduce the notion of convexity, and discuss how to optimally solve this particular problem analytically.

In *optimal decision making*, the same analogs exist. To solve a sequential decision making problem, such as when and how much to charge or discharge an electrical battery given a price signal, it is possible to repeatedly sample the solution space (in a brute force manner) to come up with a set of candidate solutions. Here, the solution space is a vector, and each element in this vector represents the control action at a particular time index. These solutions

can likewise be quickly evaluated to determine how well they perform, and the best solution can be selected. This approach has the benefit that it provides an intuitive exploration of how to formulate an optimization problem formally without getting the students bogged down in algorithmic complexities. However, as before, the wastefulness of this approach should become quickly obvious. Next, we introduce gradient free optimizers as a potential solution to speed up discovery of optimal solutions. While these should easily outperform brute force methods, their limitations also become visible to students with increasing complexity of the problem to be solved (e.g. by increasing the time horizon of the optimization problem or addition of new constraints etc.). Finally, students are introduced to convex optimization since the problem under consideration is convex. This allows students to solve the problem exactly at a far smaller computational footprint (see Fig. 1 for an illustration).

These two examples are meant to highlight the interconnection of learning and optimization theory to engineering students that may have had a background in control. Furthermore, the same approach can also be expanded to other relevant, important topics such as hyperparameter optimization etc. This approach was preferred over introducing students to a large number of machine learning and optimization algorithms, because of our experiences in the field, where even experienced energy practitioners and software developers have trouble leveraging structure in problems and choosing the right tool for the problem at hand. More concretely, over the years, we have seen software engineering colleagues, without a background in learning or optimization theory, applying (variants of) brute force search to determine optimal control actions in extremely high dimensional settings, even when the problem could be readily and exactly solved with convex optimization techniques. On the other end of the spectrum, we have seen colleagues trying to apply convex solvers to non-convex optimization problems, without fully appreciating the complexity of the challenge. The course is structured in a way to prepare students to use the right tool for the right job.

## 5. Course delivery

Beyond course content, we also explored different ways to facilitate course delivery. This was motivated by our objective to enable students without a programming background to quickly get up to speed. They were also motivated by our intention to set up a fully functional hybrid learning experience for students scattered in technical programs across Europe.

### 5.1. Using interactive notebooks

We used interactive Jupyter notebooks extensively in the teaching process to complement lectures covering theory.

Based on student feedback, this proved to be an invaluable resource in getting them up to speed. There were a number of reasons for this. Firstly, students were able to quickly apply the theoretical concepts they learned in practice. Secondly, having access to code provided a jump start of sorts and students were able to achieve a lot more than they would have otherwise. This also considerably allayed our fears of asking non-proficient programmers to read and understand existing Python code. To ensure that students could learn by doing, the notebooks also contained a number of exercises which the students had to complete themselves.

### 5.2. Using cloud infrastructure

The next question in course delivery was where to host the Jupyter notebooks, i.e. on students' workstations or the cloud. We decided to go for a cloud-based platform (Deepnote) after a thorough analysis of the advantages and disadvantages of such frameworks. The biggest benefit of such a setup is that it enables students without a programming background to focus on learning algorithms and programming, without having to figure out details such as installing libraries and setting up a functional development environment. Cloud frameworks also enable quick feedback to students as the instructor can seamlessly check in and see their code (without the need for Git commits etc.). Another advantage with such cloud-based platforms is automatic scaling of computational resources, which means students with fewer compute resources are not at a disadvantage.

### 5.3. Using programming aids

Even though the level of programming required for the course is not very high, we used a number of programming aids to help students better utilize course materials. These include Cambridge University's introductory Python programming notebooks, as well as a limited number of Datacamp licenses that students could borrow to quickly get up to speed on Python programming. Students were also given pointers to other helpful resources on programming in general and Python in particular.

## 6. Learner evaluation

The first run of the course was offered only to students at KU Leuven, and had 13 MS students from the electrical /energy engineering department. For the summer 2020 iteration, we had over 100 participants. A majority of these were students enrolled in EIT InnoEnergy MS programs, with energy practitioners and PhD researchers forming a sizable minority.

Students following these courses were evaluated on the intended learning outcomes using participation in: (1) a home-made forecasting competition on Kaggle, and (2) a

group project where the students were asked to extend what they learn in the course to apply it to a real world challenge. More concretely, the forecasting competition is meant to provide students with electricity demand data for a neighborhood over two months, which they are then asked to forecast for the next week. The choice of methodology is left to students, although they are required to explain this to their peers in a presentation. Benchmarking students' performance against predefined baselines on Kaggle also serves as an excellent way for early interventions to help struggling students. Some students only explored algorithms introduced in the lecture, while others went beyond these to also experiment with neural networks and tree-based methods. Likewise, students work in teams to complete the project where the requirement is to extend optimal decision making to also consider design choices (in the lectures, the students are only introduced to optimal control). This makes use of the same algorithms, but is conceptually harder since it interleaves learning and hierarchical optimal decision making.

## 7. Conclusions

The course, now in its second year and third iteration, is meant to introduce energy engineering students to data science. Many of these students indicated in a pre-course survey that they did not have any background in data science, or even programming. However, most thought it was extremely important for their future career objectives, and were therefore intrinsically motivated to learn more about data science in general, and machine learning in particular.

While the course provides students with a useful introduction to the broad field of data science encompassing exploratory data analysis, machine learning and optimal decision making, it is still only a high level overview. One of the next steps for the working group is to harmonize this introductory course across universities, develop follow-ups on more advanced topics, and link the course content to the decarbonization objectives that lie at the heart of this effort.

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
