# OpenReview forum: "Teaching machine learning through end-to-end decision making"
_ecmlpkdd.org/ECMLPKDD/2021/Workshop/TeachML — TeachML 2021_

### Official Review · Reviewer_xg1W · 2021-07-09
**New perspective on teaching ML to engineering students in the context of end-to-end decision making using real-world data**

**Rating:** 7
**Confidence:** 5

**Review:**

The paper presents an innovative course that provides students with an introduction to ML in the context of end-to-end decision making.

Pros:
+ Authors present an innovative course in their department.
+ Coursework problems analyze real-world data, not just toy data.
+ Discusses how ML adds value to end-to-end decision-making.
+ Presents how engineering students can learn ML.
+ Includes a course evaluation/student feedback.

Cons:
- Text does not refer to Figure 1.
- Does not discuss how the courses would be offered to students during the COVID-19 pandemic and how the course could permanently include digital learning components.

---

### Official Review · Reviewer_MsPy · 2021-07-12
**Introductory ML Course for Energy Students**

**Rating:** 6
**Confidence:** 5

**Review:**

PRELIMINARY NOTE: Anonymity is not fully respected. Institution (EIT InnoEnergy) and course ("Data science for energy students") are provided. Names of authors might be recoverable (course seems to be "locked" for the moment on their website, but teachers might be the same as another of their course: https://professionallearning.innoenergy.com/learningplan?id=6)

   The paper describes a course about ML/Data Science for Energy Students. Challenge was that students are limited in programming and software engineering background.

   Pros:
   * Skills/goal-oriented approach: In the end, students will know DS process from A to Z: EDA / ML Algos / Optimization / Real World Application / Decision Making (only lacks deployment step).
   * Course encompasses some of the challenges of DS and goes beyond technical-only knowledge: anomalous data, concept drift, statistical significance, establishment of baselines.
   * Link with optimization problems is interesting: for training and for hyperparameters tuning

   Cons:
   * Apart the description of the course content, I did not read any actual "Key Findings and Lessons Learned", apart that the course had to be case-based oriented/based on real applications
   * There's no hint on own much time is devoted to this course (3 ECTS <=> 30 hours?). How are these hours distributed on the five lessons + 4 practice sessions?
   * It seems that students where provided Jupyter Notebooks with the code. So what did they produce themselves? It is well known that doing by oneself teaches more & better than looking at what others did (https://en.wikipedia.org/wiki/Learning_pyramid).
   * Conclusion lists some "future work" ("still only a high level overview", "harmonize [...] across universities", "develop follow-ups on more advanced topics"). Would be nice to have some hints on how the authors aim to solve these problems.
   * It is said that Deepnote is better than other cloud systems as it is "non-proprietary". From what I've seen, you need to register with Google/GitHub account to access DeepNote. So how is it non-proprietary?

   A more general/philosophical question (but not less interesting): it is not clear to me how teaching students DS (i.e. asking them to build ML/DL models that we know consume a LOT of energy) will solve the problem of "anthropogenic climate change", i.e. that the fact that humans consume too much energy for too many years...

---

### Decision · Program_Chairs · 2021-07-21

**Decision:**

Accept

**Comment:**

Congratulations! The reviewers agree that this paper should be accepted.

Camera-ready version is due August 18, 2021. As you prepare the camera ready version, please take the reviewers comments into consideration.

We look forward to your participation at the workshop on September 13, 2021. We invite you also to join us for the satellite event on September 08, 2021. Schedules for both the workshop and the satellite event will be forthcoming.